# Pooled prevalence and co-occurrence of diarrhea and wasting and its associated factors among children aged 6–24 months in East Africa: Insight from recent demographic health survey: A multilevel analysis

**Alemakef Wagnew Melesse**[ID]*, **Mahlet Alehegn, Tigabu Kidie**

Department of Epidemiology and Biostatistics, Institute of Public Health, College of Medicine and Health Sciences, University of Gondar, Gondar, Ethiopia

* alemakefwagnew16@gmail.com

## Abstract

### Background

Young children experiencing both diarrhea and wasting (DW) are at increased risk of severe malnutrition, impaired immune function, and greater susceptibility to infections. This dual burden of illness significantly impacts their health, survival, and development. Diarrhea and wasting are often interlinked, with one exacerbating the other, leading to various cycle of malnutrition and frequent infections.in the long term, these conditions contribute to stunted growth delayed cognitive development and decreased survival rates. Given the vulnerability of children aged 6–24 months, addressing the dual burden of diarrhea and wasting is crucial for improving child health outcomes. This study aims to investigate the pooled prevalence and associated factors of diarrhea and wasting multimorbidity (DW) among children aged 6–24 months in East Africa.

### Methods

A pooled prevalence analysis was conducted using data from the Demographic and Health Surveys (DHS) across 11 East African countries, including a total weighted sample of 78,982 children aged 6–24 months. Given hierarchical structure of the DHS data, a multilevel binary logistics regression model was employed to identify significant factors associated with DWM. The Intra-class correlation (ICC), Median odds ratio (MOR), and LogliklihoodRatio (LLR), AIC and Deviance test were used to compare model fit and assess the contribution of different levels of variability. In bivariate analysis factors with p-value<0.2 were selected for inclusion in the multivariable multilevel logistics regression model. The final model provided the Adjusted Odds Ratios (AOR) with Corresponding 95% Confidence Intervals (CIs) to indicate the strength and directions of significance associations.

**Data availability statement:** The Demographic and Health Survey (DHS) data set is available at (http://www.dhsprogram.com). The DHS Program is authorized to distribute, at no cost, unrestricted survey data files for legitimate academic research. Registration is a prerequisite for access to data. The data sets are publicly available to all registered users and can be downloaded from the website.

**Funding:** The author(s) received no specific funding for this work.

**Competing interests:** The authors have declared that no computing interest exist.

**Abbreviations:** AIC: Akaike Information Criterion; AOR: Adjusted Odds Ratio; DW: multimorbidity of diarrhea and wasting; DHS: Demographic Health Survey; EA: East Africa; EAs: Enumeration Areas; ICC: Intra class Correlation; LLR: Log likelihood Ratio; MOR: Median Odds Ratio.

## Result

The pooled prevalence of diarrhea and wasting co-occurrence (DW) among children aged 6−24 months in East Africa was 11% (95% CI: 10.8%−11.2%), with country-specific prevalence ranging from 3% in Zimbabwe to 17% in Malawi. Several factors were significantly associated with increased odds of DWM. Male children had an 84% higher risk of DWM (AOR 1.84; 95% CI: 1.69–2.02). Children born to mothers with no formal education (AOR 3.33; 95% CI: 2.30–4.81) or with primary (AOR 2.12; 95% CI: 1.59–2.74) or secondary education (AOR 1.47; 95% CI: 1.16–3.03) also had increased odds of DWM. Inadequate sanitation facilities, such as unimproved latrine facilities (AOR 1.47; 95% CI: 1.31–1.65) and unimproved water sources (AOR 1.47; 95% CI: 1.31–1.64), were similarly associated with higher odds of DWM. Additionally, not initiating breastfeeding in a timely manner was linked to increased odds of DWM (AOR 1.25; 95% CI: 1.10–1.41). Conversely, certain factors were associated with a reduced risk of DWM. Children living in rural areas had lower odds of DWM (AOR 0.85; 95% CI: 0.68–0.94). Additionally, children residing in Mozambique (AOR 0.59; 95% CI: 0.45–0.75) and Rwanda (AOR 0.57; 95% CI: 0.42–0.77) were less likely to experience DWM.

## Conclusion

The study highlighted that the co-occurrence of diarrhea and wasting among children aged 6–24 months represents a significant public health issue in East African countries. To address this challenge, public health interventions focusing on improving maternal education, enhancing sanitation facilities, and promoting timely breastfeeding practices are essential for reducing the burden of diarrhea and wasting multimorbidity.

## Background

The co-occurrence of diarrhea and wasting (DW) represents a critical syndemic vulnerability among children under-five, in this situation, infectious disease and acute malnutrition works together in a harmful cycle that increases mortality risk [1–3]. this study focus on DW as two conditions happening at the same time, not traditional multimorbidity [4–6].

Young children face high nutritional needs during growth. At the same time, their immune systems are not fully developed. This makes them easy targets for infections like diarrhea and poor weight gain leading to wasting. In East Africa, DW contributes substantially to under-five mortality, yet it remains understudied compared to isolated conditions. [7].

DW severely impairs child health outcomes, limiting physical activities, growth, and developmental millstones while increasing hospitalization risks. Affected children require integrated management addressing both infection and nutrition, staring health care systems and families financially and emotionally [2,8].

Diarrhea causes wasting by blocking nutrient absorption in the gut. It also leads to loss of fluids, salts, and food through loose stools. On the other side, wasting makes diarrhea last longer and harder to treat [1].

Wasting weakness the body's defenses in key ways. It lowers gut immunity and changes T-cell function. Cytokine production drops, making it hard to fight germs in the intestines. Pathogens then grow more easily [1,2].

Nutritional deficiencies contribute to 45% of child death, with highest burdens in Sub-Saharan Africa. Prevalence data on DW co-occurrence vary regionally: up to 28% in Bangladesh (two-week recall), 7.2% in India, and 1.2–24.8% across sub-Saharan Africa [5,9].

Past research studies diarrhea and wasting as a separate issues. Few looked at their combined effects [1,10,11]. This study fills that gap by estimating pooled prevalence and associated factors of DW co-occurrence among children aged 6–24 months in East Africa, informing targeted interventions.

## Method

### Data source

This study used the most recent Demographic and Health Surveys (DHS) conducted after 2016–2023 from 11 East African countries (Ethiopia, Burundi, Uganda, Rwanda, Tanzania, Mozambique, Madagascar, Kenya, Zambia, and Malawi). These nationally representative datasets were merged to estimate pooled prevalence and factors of diarrhea and wasting co-occurrence (DW) among children age from 6–24months in East Africa.. Data accessed from dhsprogram.com/data/available-datasets.cfm. The child dataset(KR file) includes all live births in the five years preceding each survey. Two-stage cluster sampling used primary sampling units (enumeration Areas) based on recent census frames. Initial pooled sample: 112,456 children aged 6–24 months (Fig 1).

### Variables

Outcome: DW co-occurrence = diarrhea (≥3 loose stools past 2 weeks, per WHO/DHS) AND wasting (WHZ ≤ −2, WHO 2006 standards), coded 1 = yes, 0 = no.

**Individual factors:** child age(6–11months, 12–17 months and 18–23 months), birth weight(small, Average, large), sex-(male/female), Rota vaccination status(yes/no), mother's education(no education/primary/Secondary/higher), mother's age(15–24/25–34/35–49), mother's occupation(not working/working), perceived distance to health care(not a big-problem/problem), media exposure(yes/no), household wealth status(poorest/poorer/middle/richer), family size(5and less/6 and more), household water source(improved/not improved), household latrine access(improved/not/improved), number of under-five children(less than two/more than two), timely initiation of breastfeeding(no/yes), and birth order(first/2–3/4–5/6 and above) twin(no/ yes).

**Community- factors:** Residence(urban/rural), and country(11 categories).

### Operational definition

Media exposure: was generated by combining three variables from the survey: listening to the radio, watching television, and reading newspapers. It was categorized as "having Media exposure" (yes) if the mother had been exposed to at least one of the three sources, and "not exposed" if she had no exposure to any of the three media sources [12].

Birth weight: was categorized in to three groups. If child birth weight was less than 2,500 grams, it was classified as "small birth weight"., if birth weight was between 2,500 gram and 4,500 gram it was categorized as "average" (normal) and if birth weight is exceeded 4,500 grams, it was classified as" large birth weight" [13].

Timely initiation of breastfeeding: is defined as the initiation of breastfeeding within the first hour after birth. Children who were breastfeed within this hour were categorized as "yes" for timely initiation of breastfeeding, while those were not were categorized as "no" [14,15].

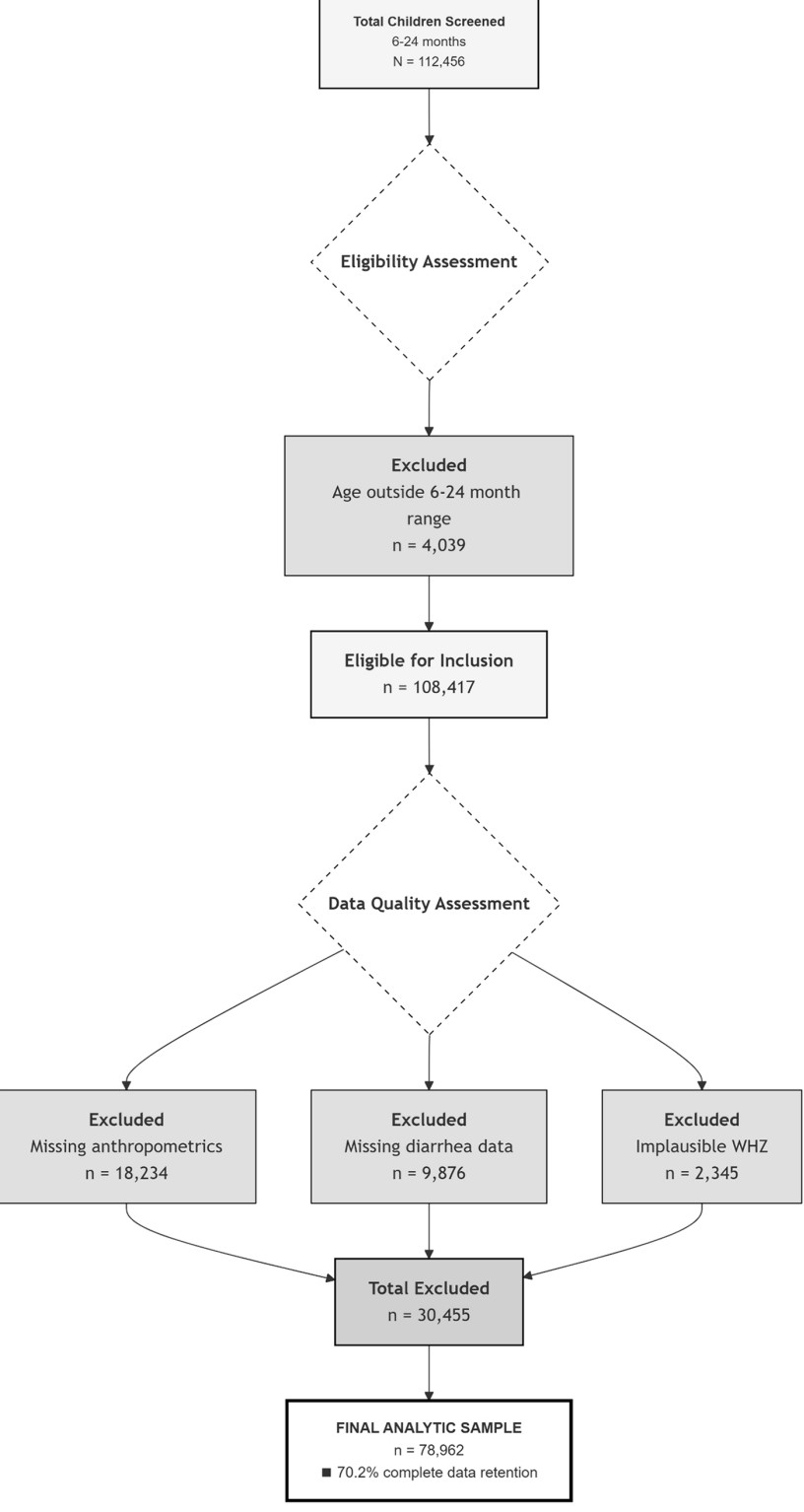

**Fig 1. Flowchart of participant selection and data processing for pooled analysis of diarrhea and wasting among children aged 6-24 months in East Africa.**

## Data management and analysis

Data from 11 east African countries were extracted cleaned and pooled in Stata 17.relevant variables selected based on literature and DHS standards. Weight is applied via svyset command accounting for sampling weight, primary sampling unit(PSU), and strata to ensure national representativeness and correct standard errors. Data quality checks include: range verification(child age 6–224 months) logical consistency(birth weight>0 kg), and outlier removal(WHZ|z|>6 SD). Exploratory analysis summarized descriptive (means, frequencies)by DW status. Pooled data prevalence calculated and visualized in forest plot by country. Missing data proportions tabulated (Table 1); complete case analysis primary, with multiple imputation(mi impute chained, m=20) sensitivity test confirming robustness(AOR shifts<3%).

## Multilevel analysis

Given DHS hierarchical structure (children nested within household/enumeration areas/countries), Multilevel mixed effects binary logistic regression modeled DW to account for intra-cluster correlation and partition variance across levels. Four sequential models were fitted;

- the null model,: assessed baseline cluster variance (δ), which contains no explanatory variables to assess the variability of DW,

- Model 1: Individual factors only. Which includes individual-level variables only,

- Model 2: Community-level factors only. Which includes community-level variables only,

- Model 3(full): both levels combined. This includes both individual and community-level factors simultaneously.

Model compared using Likelihood Ratio (LLR), deviance (−2LLR) and Akaike information criteria (AIC) lowest values of these parameters was selected as the best-fitting model [16].

## Parameter estimation method

The fixed effects measure of association was used to estimate the relationship between the likelihood of DW co-occurrence and predictor variables. Bivariable multilevel analysis screened variables((P-value<0.2 threshold) significant predictors(p<0.05) reported from multivariable model. Multicollinearity among the independent variables was assessed using the Variance Inflation Factor (VIF) <10 [17]. The logistic model equations:

$$\text{Log}\left(\frac{\pi_{ij}}{1 - \pi_{ij}}\right) = \beta_o + \beta_i x_{ij} + u_j$$

Where: $\pi_{ij}$ is the probability of the Diarrhea and wasting co-occurrence (DW), 1- $\pi_{ij}$ is the probability of not DW co-occurrence, $\beta_o$ is intercept that is the effect on DW co-occurrence when all explanatory variables are absent, $\beta_i$ is fixed coefficient $X_{ij}$ predictors for child i in cluster j, $u_j$~N (0,δ) random effect. Robust standard errors used. Multicollinearity checked via VIF (<10 all variables).

The Intra-cluster correlation coefficient (ICC) and median odds ratio (MOR) were computed to assess variability among clusters. The ICC quantifies the degree of heterogeneity of the outcome between clusters, representing the proportion of the total variation in DW attributable to between cluster variations. The ICC takes values between 0 and 1, where a higher value indicates greater variability among clusters. Therefore, this implies that a multilevel model is necessary for this specific dataset. It also indicates how much of the variations in the response are explained by clustering. The ICC is calculated as:

$$ICC = \frac{VA}{(V_A + \pi 2/3)}$$

**Table 1.** Individual and community level characteristics of the study population in East Africa for co-occurrence of diarrhea and wasting.

| Variables | Categories | Weighted frequency | Weighted % |
|---|---|---|---|
| **Child age** | 6-11 | 8,111 | 10.27 |
| | 12-17 | 63,557 | 80.49 |
| | 18-23 | 7,294 | 9.24 |
| **Child sex** | Male | 39,920 | 50.56 |
| | Female | 39,042 | 49.44 |
| **Number of under five children** | ≤ 2 | 67,379 | 85.33 |
| | ≥ 3 | 11,583 | 14.67 |
| **Mother age** | 15-24 | 26,387 | 33.42 |
| | 25-34 | 35,806 | 45.35 |
| | 35-49 | 16,769 | 21.24 |
| **Mother's occupation** | Not working | 23,550 | 29.82 |
| | Working | 55,412 | 70.18 |
| **Mother's Educational status** | No education | 16,694 | 21.14 |
| | Primary level | 41,145 | 52.11 |
| | Secondary level | 17,894 | 22.66 |
| | Higher | 3,229 | 4.09 |
| **Toilet facility** | Unimproved | 25,939 | 32.85 |
| | Open | 11,120 | 14.08 |
| | Improved | 41,903 | 53.07 |
| **Water source** | Unimproved | 21,640 | 27.41 |
| | Improved | 57,322 | 72.59 |
| **Wealth status** | Poorest | 18,832 | 23.85 |
| | Poorer | 16,577 | 20.99 |
| | Middle | 15,230 | 19.29 |
| | Richer | 14,829 | 18.78 |
| | Richest | 13,494 | 17.09 |
| **Birth weight** | Small 21,820 27.63 | | |
| | Average | 35,853 | 45.41 |
| | Large | 21,289 | 26.96 |
| **Birth order** | First | 19,621 | 24.85 |
| | 2-3 | 29,092 | 36.84 |
| | 4-5 | 16,742 | 21.20 |
| | 6 and above | 13,507 | 17.11 |
| **Distance to health facility** | Not big problem | 44,768 | 56.70 |
| | Big problem | 34,194 | 43.30 |
| **Media accesses** | No | 32,969 | 41.75 |
| | Yes | 45,993 | 58.25 |
| **Rota virus vaccine** | No | 13,305 | 16.85 |
| | Yes | 65,657 | 83.15 |
| **Family size** | 5 and less | 41,408 | 55.44 |
| | 6 and above | 37,554 | 47.56 |
| **Twin** | No | 76,247 | 96.56 |
| | Yes | 2,715 | 3.44 |
| **Timely initiation of breast feeding** | No | 10,898 | 13.80 |
| | Yes | 68,962 | 86.20 |

*(Continued)*

**Table 1.** (Continued)

| Variables | Categories | Weighted frequency | Weighted % |
|---|---|---|---|
| Residence | Urban | 16,491 | 20.88 |
| | Rural | 62,471 | 79.12 |
| Country | Burundi | 9,018 | 11.46 |
| | Ethiopia | 3,751 | 4.76 |
| | Kenya | 7,148 | 9.08 |
| | Madagascar | 8,303 | 10.55 |
| | Malawi | 13,375 | 16.97 |
| | Mozambique | 6,425 | 8.16 |
| | Rwanda | 5,432 | 6.90 |
| | Tanzania | 7,263 | 9.23 |
| | Uganda | 12,111 | 15.38 |
| | Zambia | 3,516 | 4.47 |
| | Zimbabwe | 2,396 | 3.04 |

Where: $V_A$ is cluster level variance.

The Median Odds Ratio (MOR) is defined as the median value of the odds ratio between the area with the highest risk of Diarrhea and wasting multimorbidity and the area with the lowest risk when randomly selecting two clusters. The MOR provides a measure of variability in the odds of experiencing DW across different clusters. Mathematically, the MOR can be expressed as:

$$MOR = e^{\sqrt{2 \times \sigma^2 \times 0.6745}} \; Or \; MOR = e^{0.95\sqrt{\sigma^2}}$$

Where: δ is the area level variance.

A higher MOR indicates greater disparity in the odds of DW between clusters, highlighting the importance of community-level factors in influencing health outcomes. This measure reinforces the need for multilevel modeling to accurately capture the effects of both individual and community characteristics on the likelihood of diarrhea and wasting (DW).

### Ethics approval and consent to participate

This study is a secondary data analysis of previous collected and published household survey data. As such, it did not require independent ethical approval. However, the ethical approval and permission to access the data were obtained from the DHS website https://www.measuredhs.com. All DHS surveys were approved by ICF international and an institution review Board (IRB) in each country, in accordance with the united states department of health and Human Services requirements for human subject protection and the ethical standards are available at http://goo.gl/ny8T6X. All methods were carried out according to the Declaration of Helsinki.

## Results

### Characteristics of study participants

A total of 78,692 children aged 6–24 months were included in the final analysis, among these, 63,557(80.5%) from rural areas(79.1%).were in the age group of 12–17 months. Of the participants, 39,920 (50.49%) were male and 83.15% had received full Rota virus vaccination. The majority of the children, 76,247 (96.62%) were born as single births, while 16.97% of children resided in Malawi (Table 1).

**Prevalence of diarrhea and wasting co-occurrence among children aged from 6–24 months in East Africa**

The forest plot reveals heterogeneity in prevalence estimates across countries, with Malawi showing the highest prevalence (17.0%, 95% CI: 16.3–17.7) and Zimbabwe the lowest (2.8%, 95% CI: 2.2–3.5). The non-overlapping confidence intervals between extreme estimates (e.g., Malawi vs. Zimbabwe) suggest statistically significant differences in prevalence between countries with the highest and lowest burden.

The pooled prevalence of co-occurrence of diarrhea and wasting (DW) among children aged from 6−24 in East Africa was 11% (95% CI: 10.8% − 11.8%). This prevalence varied significantly across countries; with the highest rate observed in Malawi at 17% (CI:16%−18%) and the lowest rate in Zimbabwe at 3%(CI: 2%−4%) (Fig 2 and Fig 3).

**Fig 2. Prevalence of Diarrhea and wasting co-occurrence among children aged from 6-24 months in East Africa 2016-2024.**

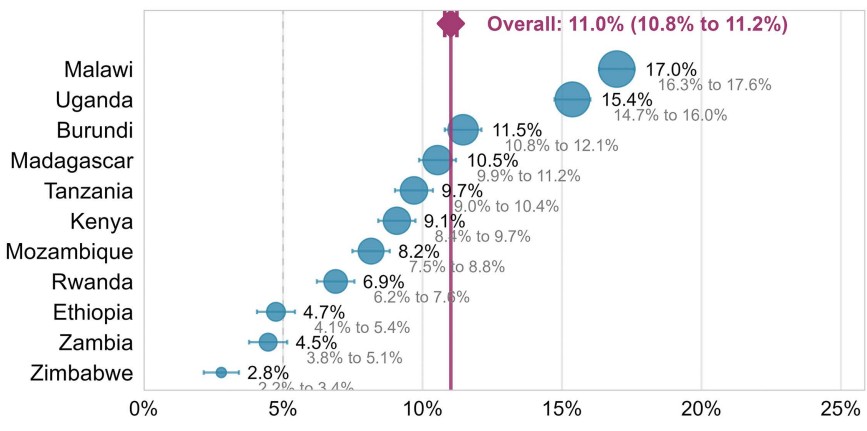

Data from 11 countries (Total N = 78,962 children)

Note: Point size represents each country's contribution to the total sample size.

**Fig 3. Pooled Prevalence of Diarrhea and wasting co-occurrence among children aged from 6-24 months in East Africa 2016-2024.**

## Random effect model and model fitness

In the null model, ICC = 20.1% indicated substantial between-cluster variation in DW. MOR = 2.39 showed that when randomly selecting two children with identical characteristics from different clusters, the child from the higher-risk cluster had 2.39 times higher odds of DW. After including predictors, MOR decreased to 1.87, indicating the full model explained much of this between-cluster variability (Table 2). Model III (full model) showed best fit with lowest AIC (19,223) and deviance. (Table 2)

## Associated factors of DW co-occurrence among children aged from 6–24 months

In the multivarite multilevel binary logistics regression Analsis; several factors were significantly associated with increased odds of diarrhea and wasting(DW) co-occurrence among children aged 6t o 24 months.This factors included Maternal education, Age of children, Sex, number of underfive children, birth weight, rota virus vaccination, distance to health facility, media exposure, water source, laterine, and breast feading practices.Conversly, maternal Age and residence

**Table 2. Parameters and model fit statistics for multilevel regression analysis models.**

| Parameters | Null model | Model I | Model II | Model III |
|---|---|---|---|---|
| Variance | 0.848 | 0.807 | 0.844 | 0.806 |
| ICC | 20.05% | 19.7% | 18.19% | 17.7% |
| MOR | 2.39 | 2.35 | 2.25 | 2.23 |
| Model fitness | | | | |
| Log like-hood | −10326.585 | −9895.6589 | −10318.105 | −9894.9146 |
| Deviance | 20,653.17 | 19,791.318 | 20,636.11 | 19,789.829 |
| VIF | -------- | 2.66 | 1.38 | 2.75 |
| AIC | 20033.53 | 19337.55 | 19866.27 | 19223.73 |

ICC = Inter cluster correlation coefficients, VIF= Variance Inflation Factor AIC= Akaike information criteria MOR = Median odds ratio

significantly decrease the odds of DW multimorbidity among this age group. Specially, Children aged 6–11 and 12–17 months had Adjusted Odds ratio of 1.49:95%CI(1.16–1.91) and 2.32(95%CI:1.89–2.86), respectively, indicating higher odds of experiencing DW compared to children aged 18–24 months. Being Male increased the odds of DW multi-morbidity by 84% (95%CI: 1.69–2.02) relative to female children. Additionally, households with three or more under-five children were found to have 1.42 times (95% CI: 1.25–1.60) higher risk of DW compared to households with two or fewer under-five children. Children who were Underweight had a 1.2 times (95% CI: 1.08–1.34) higher risk of DW compared to those normal and above-normal birth weight. Not receiving the Rotavirus vaccine was associated with 1.6 times (95% CI: 1.53–1.68) higher risk of DW compared to vaccinated children. Furthermore, children without timely initiation of breast feeding had a 1.25times (95% CI: 1.1–1.41) higher risk of DW to those who were breastfed on time. Regarding maternal age, mothers aged 15–24 years and 25–34 years had 34% (AOR:0.66 CI:0.55–0.79) and 22% (AOR 0.78 with CI:0.68–0.89) lower risk of DW, respectively, compared to maternal aged 35–49 years(Table 3).

From community-level perspective, rural residence was associated with a 20%(AOR:0.85, CI:0.68–0.94) lower risk of DW co-occurrence compared urban residence. Children living in Mozambique and Rwanda had 31% and 43% lower risk of DW, respectively,compared with children from Uganda. Incontrast, children from Burundi and Zimbabwe had 2.3 and 1.7 times more risk of DW compared to children from Uganda(Tbale 3.).

## Discussion

This study, reveals a 11% pooled prevalence of diarrhea-wasting co-occurrence(DW) among East African children aged 6–24 months, representing over 2 million affected children annually given regional under-five population [2,18,19]This syndemic burden exceeds isolated diarrhea(14–15% or wasting (7–9%) prevalence, highlighting multiplicative mortality risk from bidirectional interaction[UNICEF]. Country variation (3–17%) underscores need for targeted interventions beyond uniform regional strategies.

Younger children(6–17 Months) showed 1.5–2.3 times higher DW odds versus 18–24months, consistent with Ethiopian and Saudi studies [20–22].This reflects peak weaning vulnerability when complementary feeding confides with crawling/ exploration increasing enteric pathogen exposure, combined with immature immunity unable to maintain gut barrier integrity during diarrheal episodes.

Children of younger mothers(15–34years) had 22–34% lower DW odds versus mothers ≥35 years, persisting after stratification by education/ wealth [1,22]. This contrasts typical young-mother risk profiles but aligns with DHS patterns showing higher antenatal care(89% vs 76%) and institutional delivery(72% vs64%)among younger mothers. Survival bias may also contribute: Higher children of older mothers more likely to reach 6–24 months deposited early risks.

Rural residence showed 15% lower odds(AOR = 0.85), contradicting urban advantage hypothesis. DHS data reveals rural advantages in exclusive breastfeeding (OR=1.34) and lower formula feeding(12% vs 28% urban) [23]. Stronger kinship networking reduce orphan hood (3.1% vs 6.8%)and support caregiving during illness. These sociocultural protections appear to outweighs urban WASH advantages in this age group [24]. Households with ≥3 under-5 children had 42% higher DW odds, matching Ethiopian findings [25–27]. Overcrowding facilitates fecal-oral pathogen transmission while diluting caregiving attention and nutrition allocation. This underscores need for child spacing programs alongside WASH in high-fertility contexts [28].

Unimproved water/latrine doubled DW risk (AOR=1.47, 95%CI: 1.31–1.65), while health facility distance problems increased odds 10% (AOR=1.10, 95%CI: 1.01–1.22). These findings confirm persistent environmental and structural barriers to child health. Fecal-oral transmission through contaminated water drives acute diarrhea precipitating wasting, while access barriers limit timely rehydration and growth monitoring. Integrated WASH-nutrition platforms delivered through community health workers represent proven scalable solutions for these modifiable risks [29,30]

Country effects revealed stark disparities: Burundi (AOR=2.37) and Malawi (17% prevalence) vs Rwanda (AOR=0.57, 5% prevalence). High-burden countries show lower Rota coverage (Malawi 72%, Burundi 68% vs Rwanda 91%), weaker

**Table 3.** Multivariable multilevel logistic regression analysis for factors associated with DW co-occurrence in East africa 2016-2024.

| Variables | Null model (empty) | Model 1 | Model 2 | Model 3 |
|---|---|---|---|---|
| **Age of child** | | | | |
| 6-11 | | 1.47(1.14-1.89)* | | 1.49(1.16-1.91)* |
| 12-17 | | 2.36(1.92-2.89)** | | 2.32(1.89-2.86)** |
| 18-24 | | 1 | | 1 |
| **Sex of child** | | | | |
| Female | | 1 | | 1 |
| Male | | 1.83(1.68-2.00)** | | 1.84(1.69-2.02)** |
| **Number of children under the age of five** | | | | |
| 2 and less | | 1 | | 1 |
| 3 and above | | 1.43(1.26-1.61)** | | 1.42(1.25-1.60)** |
| **Birth weight** | | | | |
| Small | | 1.48 (1.31, 1.67)** | | 1.20(1.08-1.34)** |
| Large | | 0.88 (0.77-1.001) | | 0.82(0.73-1.02) |
| Average | | 1 | | 1 |
| **Mother's educational status** | | | | |
| No education | | 4.07(2.84-5.83)** | | 3.33(2.3-4.81)** |
| Primary | | 2.37(1.67-3.36)** | | 2.12(0.59-0.74)** |
| Secondary | | 1.61(1.13-2.29)* | | 1.49(1.46-3.03)* |
| Higher | | 1 | | 1 |
| **Mother's age** | | | | |
| 15-24 | | 0.68(0.57-0.81) ** | | 0.66(0.55-0.79)** |
| 25-34 | | 0.82(0.72-0.93) * | | 0.78(0.68-0.89)** |
| 35-49 | | 1 | | 1 |
| **Latrine** | | | | |
| Unimproved | | 1.44(1.29-1.6)** | | 1.47(1.31-1.65)** |
| Open | | 1.11(0.65-1.00) | | 1.31(1.12-1.55)* |
| Improved | | 1 | | 1 |
| **Water source** | | | | |
| Unimproved | | 1.31(1.18-1.46)* | | 1.47(1.31-1.65)** |
| Improved | | 1 | | 1 |
| **Distance to health facility** | | | | |
| Not big problem | | 1 | | 1 |
| A big problem | | 1.08(0.98-1.1) | | 1.10(1.008-1.22)* |
| **Household wealth index** | | | | |
| Poorest | | 0.87(0.72-1.05) | | 1.04(0.87-1.24) |
| Poorer | | 0.80(0.67-0.96) | | 0.91(0.74-1.17) |
| Average | | 0.85(0.71-1.02) | | 0.97(0.80-1.17) |
| Richer | | 0.96(0.81-1.13) | | 1.04(0.87-1.24) |
| Richest | | 1 | | 1 |
| **Twin** | | | | |
| No | | 1 | | 1 |
| Yes | | 0.94(0.85-1.05) | | 0.97(0.87-1.09) |
| **Timely initiation of breastfeeding** | | | | |
| No | | 1.35(1.20-1.52)** | | 1.25(1.1-1.41)** |
| Yes | | 1 | | 1 |

*(Continued)*

Table 3. (Continued)

| Variables | Null model (empty) | Model 1 | Model 2 | Model 3 |
|---|---|---|---|---|
| **Media** | | | | |
| **No** | | 1.1(1.003-1.21)* | | 1.14(1.04-1.26)* |
| **Yes** | | 1 | | 1 |
| **ROTA** | | | | |
| **Yes** | | 1 | | 1 |
| **No** | | 1.64(1.51-1.72)** | | 1.607(1.53-1.68)** |
| **Residence** | | | | |
| **Urban** | | | 1 | 1 |
| **Rural** | | | 1.17(1.02-1.34)* | 0.85(0.68-0.94)* |
| **Country** | | | | |
| **Uganda** | | | 1 | 1 |
| **Burundi** | | | 0.02(0.018-0.26)** | 2.37(1.94-2.89)** |
| **Ethiopia** | | | 0.88(0.68-1.15) | 0.98(0.73-1.29) |
| **Kenya** | | | 0.51(0.4-0.65)** | 0.86(0.66-1.129) |
| **Madagascar** | | | 0.81(0.66-0.99)* | 0.89(0.71-1.11) |
| **Malawi** | | | 0.69(0.57-0.83)** | 1.21(0.98-1.50) |
| **Mozambique** | | | 0.61(0.48-0.78) ** | 0.59(0.45-0.75)** |
| **Rwanda** | | | 0.37(0.28-0.49)** | 0.57(0.42-0.77)** |
| **Tanzania** | | | 0.88(0.71-1.08) | 1.14(0.91-1.46) |
| **Zambia** | | | 0.59(0.44-0.79)** | 1.09(0.83-1.44) |
| **Zimbabwe** | | | 0.84(0.62-1.13) | 1.7(1.9-2.89)** |

[a]Reference category. *p < 0.001. COR = crude odds ratio; AOR = adjusted odds ratio

nutrition policies (index scores 54/100 vs 78/100), and reduced WASH budgets (0.8% vs 1.4% GDP). Rwanda's community health insurance (92% coverage) and results-based financing contrast Malawi's fragmented systems. These macro-differences explain 23% between-country variance beyond individual factors [31,32].

The weighted pooled prevalence of diarrhea and wasting co-occurrence (DW) among children aged 6–24 months across East Africa(EA) was found to be 11% (CI: 10.8% − 11.2%). The highest prevalence was observed in Malawi (17% 95% CI: 16–18%), while the lowest was in Zimbabwe (3 CI: 2–4%). This finding align with previously studies [33,34], reinforcing the significance of DW as public health issue in the region.

After controlling for clustering effects, the multilevel analysis revealed that children aged 6–11 and 12–17 months had 1.49 and 2.3 times higher odds of experiencing DW compared to those aged 18–24 months. This observation is consistent with studies conducted in southwestern Saudi Arabia and Ethiopia [35–37]. A potential explanation for this trend is that older children are more likely to have better immune system, which may better equip them to combat infections, thus offering greater protection against wasting.

This study found that children with low and average birth weights were associated with higher odds of Diarrhea and wasting multimorbidity (DW) compared to those with larger birth weights. A plausible explanation for this is that low birth weighted infants often have underdeveloped immune systems, Making them more susceptible to recurrent infections and poor weight gain. This finding is consistent with research conducted in Pakistan [38,39].

Additionally, our study indicated that lower maternal age was negatively associated with DW compared to older maternal age. While, some literature [12,40] suggests that younger maternal age as a risk factor of DW, our finding revealed

that the odds of DW among children born to mothers aged 15–24 and 25–34 had 34% and 22% lower odds of DW, respectively, compared to those born to mothers aged 35–48. This counterintuitive result highlights the complexity of maternal influences on child health and suggests that factors such as maternal health and socioeconomic status may play a significant role in these outcomes [41].

The number of children under the age of five living in a household emerged as another significant factor associated with diarrhea and wasting multimorbidity (DW). Our findings indicate that having three or more children under-five years of age is positively correlated with increased odds of DW. This result aligns with previous studies conducted in Ethiopia and East Africa [12,42]. A plausible explanation for this association is highlighted risk of infections due to overcrowding, coupled with insufficient resources and a lack of parental attention in households with a larger number of young children. This conditions can hinder effective caregiving and increase vulnerability to illnesses, further exacerbating the risk of DW [11,43].

The absence of Rotavirus vaccination was identified as a significant risk factor for diarrhea and wasting multimorbidity (DW). This finding is Consistent with previous studies conducted in Ethiopia [44] and Thailand [45]. The possible explanation is that vaccinated children benefit from enhanced protection against diarrheal diseases, which are among the most common infections affecting wasted children. By reducing the frequency and severity of diarrheal episodes, rotavirus vaccination indirectly helps prevent wasting [46]. Additionally, lack of media exposure was also found a risk for DW. This result aligns with prior researches [47], suggesting that media serves as a crucial source of up-to-date information that can improve child health management. Media exposure facilitates awareness and education, which lead to behavior changes that promote better health practices [47,48].

In addition to previous mentioned factors, the distance to health facility was identified as a significant risk factor for diarrhea and wasting multimorbidity. Greater distance can hinder access to healthcare services, reducing the frequency of visits necessary for essential preventive measures such as vaccinations and educational programs on nutrition and hygiene. This barrier can lead to missed opportunities for intervention, ultimately exacerbating health complications and increased mortality risks among [49]. The finding is consistent with earlier studies that have highlighted the impact of distance on health care access, particularly concerning diarrheal diseases [10].By understanding this barriers, we can better advocate for improved health care infrastructure and accessibility, which are critical for reducing DW in vulnerable populations.

In the current study, rural residence was found to be negatively associated with diarrhea and wasting co-occurrence (DW). This finding contrasts to most previous research [50,51], but aligns with study conducted in Ghana [52]. The potential explanation for this discrepancy is that rural residents may be more inclined to engage in extended breastfeeding practices and benefit from stronger community support in child-rearing [53]. Furthermore, when comparing the association of diarrhea and wasting across different countries, Burundi and Zimbabwe exhibited a positive association, while, Mozambique and Rwanda showed a negative association relative to Uganda. These differences may be attributed to variations in health care financing, vaccination coverage, Status of children, healthcare infrastructures, community health programs, and the implementations of result based financing (RBF) system focused on maternal and child health across these nations [54,55].

## Strength and limitation of the study

Strengths include large pooled DHS sample, multilevel modeling of clustering, and comprehensive covariate adjustment. Limitations: cross-sectional design prevents causality; timely breastfeeding initiation crude measure (future studies should use exclusive/continued breastfeeding); DHS two-week diarrhea recall may miss chronic cases. Prospective cohorts with biological markers recommended.

## Conclusion and recommendation

DW co-occurrence affects nearly 1 in 11 East African children aged 6–24 months, with younger age, overcrowding, and WASH gaps as key modifiable risks. Counterintuitive rural/maternal age protections highlight sociocultural factors

requiring contextualized interventions. Country disparities demand tailored policies strengthening Rwanda-like systems while addressing Malawi/Burundi structural deficits. Integrated syndemic approaches combining vaccination, WASH, and family planning offer greatest impact.

## Acknowledgments

We greatly acknowledge Measure Demographic health survey programmers for granting access and sending the complete data set of East African Countries upon request.

## Author contributions

**Conceptualization:** Alemakef wagnew Melesse, tigabu kidie, Mahlet Alehegn.

**Data curation:** Alemakef wagnew Melesse, Mahlet Alehegn.

**Formal analysis:** Alemakef wagnew Melesse, Mahlet Alehegn.

**Funding acquisition:** Mahlet Alehegn.

**Investigation:** Alemakef wagnew Melesse, tigabu kidie, Mahlet Alehegn.

**Methodology:** Alemakef wagnew Melesse, tigabu kidie, Mahlet Alehegn.

**Project administration:** Alemakef wagnew Melesse, Mahlet Alehegn.

**Resources:** Mahlet Alehegn.

**Software:** Alemakef wagnew Melesse, Mahlet Alehegn.

**Supervision:** Alemakef wagnew Melesse, tigabu kidie.

**Validation:** Alemakef wagnew Melesse, Mahlet Alehegn.

**Visualization:** Alemakef wagnew Melesse, Mahlet Alehegn.

**Writing – original draft:** Alemakef wagnew Melesse, tigabu kidie, Mahlet Alehegn.

**Writing – review & editing:** Alemakef wagnew Melesse, tigabu kidie, Mahlet Alehegn.

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
