## [Decision Letter · Decision Letter 0]

21 Dec 2025

PONE-D-25-19278Pooled prevalence and  Multimorbidity of diarrhea and wasting and its associated factors among children aged 6-24 months in East Africa: Insight from Recent Demographic Health Survey:  A Multilevel analysisPLOS One

Dear Dr. Melesse,

Thank you for submitting your manuscript to PLOS ONE. After careful consideration, we feel that it has merit but does not fully meet PLOS ONE’s publication criteria as it currently stands. Therefore, we invite you to submit a revised version of the manuscript that addresses the points raised during the review process.

We look forward to receiving your revised manuscript.

Kind regards,

Edison Arwanire Mworozi, M.D

Academic Editor

PLOS One

**Journal Requirements:**

2. Please amend either the abstract on the online submission form (via Edit Submission) or the abstract in the manuscript so that they are identical.

3. Please upload a new copy of Figure 1 as the detail is not clear. Please follow the link for more information:  https://journals.plos.org/plosone/s/figures

4. Please include your tables as part of your main manuscript and remove the individual files. Please note that supplementary tables (should remain/ be uploaded) as separate "supporting information" files.

5. We note that there is identifying data in the Supporting Information file < Table 1_DW.docx >. Due to the inclusion of these potentially identifying data, we have removed this file from your file inventory. Prior to sharing human research participant data, authors should consult with an ethics committee to ensure data are shared in accordance with participant consent and all applicable local laws.

-Location data

Please remove or anonymize all personal information, ensure that the data shared are in accordance with participant consent, and re-upload a fully anonymized data set. Please note that spreadsheet columns with personal information must be removed and not hidden as all hidden columns will appear in the published file.

6. Please include captions for your Supporting Information files at the end of your manuscript, and update any in-text citations to match accordingly. Please see our Supporting Information guidelines for more information: http://journals.plos.org/plosone/s/supporting-information .

**Additional Editor Comments:**

Please read and comprehend the comments made by the reviewers then revise accordingly. Have a great week.

Reviewers' comments:

Reviewer's Responses to Questions

**Comments to the Author**

1. Is the manuscript technically sound, and do the data support the conclusions?

Reviewer #1: Partly

Reviewer #2: Yes

2. Has the statistical analysis been performed appropriately and rigorously? 

Reviewer #1: No

Reviewer #2: Yes

3. Have the authors made all data underlying the findings in their manuscript fully available?

Reviewer #1: Yes

Reviewer #2: Yes

4. Is the manuscript presented in an intelligible fashion and written in standard English?

Reviewer #1: No

Reviewer #2: Yes

5. Review Comments to the Author

Reviewer #1: This manuscript addresses an important and timely topic by analyzing the diarrhea and wasting among young children in East Africa using Demographic and Health Survey (DHS) data. The authors aim to estimate the pooled prevalence of this co-occurrence and identify associated factors through multilevel analysis. While the study’s focus is relevant for public health and nutrition policy in low-income settings, several conceptual, methodological, and reporting issues reduce the clarity, coherence, and scientific rigor of the manuscript. The following comments aim to provide constructive suggestions for improving the theoretical framing, variable and model explanation, and interpretation of results.

Introduction

Definition of multimorbidity

The central conceptual framework of the manuscript is problematic. The authors repeatedly frame the co-occurrence of diarrhea and wasting as a form of “multimorbidity.” However, this interpretation does not align with standard definitions in the literature, which describe multimorbidity as the co-existence of multiple chronic conditions within the same individual, typically requiring long-term care or management. Diarrhea, as it is defined a in this study, reflects an acute condition in children. Wasting is referred to as "acute malnutrition", implying a condition with clear short-term or recent onset. However, later in the text, wasting is defined as "a weight-for-height ratio more than two standard deviations below the mean for the reference population", which does not convey any temporal dimension and creates a conceptual inconsistency. Change the study as an investigation of co-occurring entities or syndemic vulnerabilities would be conceptually more accurate and scientifically defensible. the use of “multimorbidity” is misleading and weakens the study’s coherence.

Also, the manuscript lacks consistency in the use of terminology when referring to the co-occurrence of diarrhea and wasting. The authors use multiple labels—“DW,” “DWM,” “DW multimorbidity,” “diarrhea and wasting multimorbidity”— and this leads to confusion about whether “DW” is meant to represent: a descriptive epidemiological finding (as the co-occurrence of two distinct conditions), a new conceptualized clinical entity, a binary variable in the statistical model, or a form of “multimorbidity” in the traditional sense (which would be incorrect, as previously noted).

First, the manuscript should explain if DW is treated as a clinical concept, an analytic outcome, or both. Also, it should define how “DW” is constructed as a variable, using a consistent acronym and label.

Methods

Inclusion and exclusion criteria

The manuscript fails to report the inclusion and exclusion criteria used in the final analytic sample of 78,962 children. It is unclear if children without anthropometric measurements or recent diarrhea data were excluded, if implausible anthropometric values were removed, if the sample includes all children in the age range or only those with complete data… It is recommended that the authors provide a transparent flow of case inclusion (preferably as a diagram), explicitly defining all exclusion criteria and justify the final sample size in the context of the original population.

In line with this, the manuscript provides no information on how missing data were handled, either for the primary outcome (DW) or for covariates. This is an important omission. The authors should quantify the proportion of missing data for each variable used, explain the strategy chosen to manage missing values and justify the approach.

Selected variables

Although the manuscript explains the derivation of some constructed variables (media exposure, birth weight), it remains unclear how other independent variables were operationalized. The authors should explicitly state if these variables were used in their original DHS format or additional recoding was applied.

Analytic approach

The afirmation “this structure could violate the assumptions of independence of observations and equal variance inherent in traditional logistic regression models” is conceptually confused and is not a valid justification for the choice of multilevel modeling, since concerns regarding independence and variance assumptions are already addressed through the survey design correction applied via the svy commands.

The authors employed a multilevel mixed-effects binary logistic regression model to estimate the association between child-level and community-level factors and the likelihood of co-occurring diarrhea and wasting (“DW”). Multilevel modeling should be justified based on theoretical grounds and the need to model between-cluster variability explicitly. this choice could be appropriate given the aim of the study and the hierarchical nature of DHS data.

Results

The results section presents some weaknesses in reporting, interpretation and clarity.

Table 1 presents several issues that limit its clarity and interpretability. First, the total number of children included in the analysis is not reported, which is essential to contextualize the percentages. Some units of measurement for variables are not clearly indicated. Additionally, for dichotomous variables, it would improve readability to report only one category to avoid redundancy (“Child sex”,”Mother ocupation”…).

It would be helpful to report the prevalence of diarrhea and malnutrition separately, in addition to the joint outcome (DW), as this would offer greater epidemiological context and comparability with prior studies.

The interpretation of the MOR could be clarified. First, the use of the term "significant variation" could be misinterpreted in a statistical sense. To avoid ambiguity, it would be clearer to use terms such as "substantial" or "considerable" variation when interpreting the MOR. The MOR does not compare children in areas of "high vs. low prevalence", but rather quantifies the median increase in odds when comparing two children with identical characteristics from two randomly selected countries. A MOR of 2.39 reflects substantial between-country heterogeneity. A decrease in the MOR from the null model to Model III actually indicates that the model explains more, not less, of the between-cluster variability in the outcome.

The AOR for rural residence of 0.85 is inaccurately described as a 20% reduction, when it actually represents a 15% lower odds of DW. Also, there appears to be a formatting or numerical error in the reporting of the adjusted odds ratio for “Secondary” in mother education status: the AOR is reported as 1.46, but the 95% confidence interval is 1.49–3.03, which is inconsistent.

The use of a forest plot typically implies that independent estimates ( from different studies or settings) are being synthesized, often using meta-analytic methods. That is clearly not the case here.The plot assigns equal weights (9.1%) to each country, despite substantial differences in sample sizes and it is not well explained what the plotted values reflect. If the intention was to display variation in prevalence across countries, alternative formats such as geographical maps would be more appropriate and transparent. At a minimum, the figure should be accompanied by a clear methodological explanation.

Discussion

Some changes are needed to improve the clarity of the discussion.The beginning of the discussion repeat the study aim and results with little interpretive value.

The authors report a weighted pooled prevalence of diarrhea and wasting multimorbidity (DW) of 9% among children aged 6–24 months in East Africa and cite references 22 and 23 to support the significance of this burden. However, upon closer examination, these references pertain to studies on breastfeeding practices , not to the prevalence of diarrhea, wasting, or their co-occurrence. This makes them inappropriate citations in this context.

Additionally, while references 16 and 31 (Demissie et al., 2021; Tareke et al., 2022) do report on the prevalence of diarrhea alone in sub-Saharan and East African children (approximately 14–16%), they do not assess the joint occurrence of diarrhea and wasting. These sources cannot be used to validate or compare the DW prevalence reported in the current study.

The finding that younger maternal age (15–24 and 25–34 years) is associated with significantly lower odds of DW compared to older mothers (35–49 years) appears counterintuitive, especially considering previous literature suggesting that young maternal age is a risk factor for child undernutrition and illness (e.g., refs 16, 29). However, this result may be influenced by several underlying factors that are not adequately explored in the discussion.

It would strengthen the study to explore whether the effect of maternal age persists when stratifying by key mediating factors (maternal education, household wealth, access to care...)

It also happens with the association between rural residence and lower DW risk. It contradicts many prior studies. The proposed explanation (“more extended breastfeeding and community support”) is not enough substantiated with evidence.

While the study provides valuable insights, it may be worth reconsidering the selection and operationalization of some key variables. For instance, breastfeeding is only captured through the timing of initiation within the first hour of birth. As i know, DHS offers additional indicators such as exclusive breastfeeding duration, continued breastfeeding at 1 and 2 years, etc. which are strongly associated with both diarrhea and malnutrition.

Differences between countries (Malawi and Zimbabwe) are just mentioned .The authors selected a multilevel modeling approach that includes country but this choice is not matched by a sufficiently robust analytical or interpretative emphasis on country-level differences.

There is no enough consideration of potential macro-level determinants such as differences in health system infrastructure, national nutrition or vaccination programs, water and sanitation policies, or socioeconomic indicators. The authors could include a brief summary of key contextual differences between countries with notably high/low DW prevalence and discuss possible structural or policy-level explanations for these differences (using DHS data or relevant literature).

Grammatical, typographical, and syntactic errors

The manuscript contains grammatical, typographical, and syntactic errors. for example: “This studyis based...” “Children who were breastfeed...” “among children age from 6–24months” “included in to multivariable analysis”. Also, some variable labels are misspelled: “laterine” , “Media accesses” , “birth weight small”… These errors hinder readability. A language revision by an english speaker or professional editor with experience in academic writing is recommended.

General conclusions

The main issue with this article lies in its conceptual framework, which is fundamentally flawed. The outcome variable combines an acute infectious disease (diarrhea) with a non-acute condition (wasting) under the label of multimorbidity. This does not align with standard definitions of multimorbidity in the literature and undermines the theoretical coherence of the study.

The use of multilevel modeling is, in principle, an appropriate analytic strategy. However, if the model is designed to account for between-country variability, the manuscript should include a substantial section that compares and interprets the results across countries with detail. This aspect is largely underdeveloped.

The article also fails to describe how missing data were handled or what inclusion and exclusion criteria were applied in the final analytic sample.

In addition, the manuscript contains several minor issues—including grammatical and typographical errors, inconsistencies in variable labeling, and occasional formatting problems in tables. Moreover, the presentation of the results is suboptimal, with incomplete descriptive statistics, unclear figure and insufficient explanation of key tables and variables. these issues are relatively easy to correct , they remain secondary to the more fundamental concerns regarding the study’s conceptual framework and outcome definition.

Reviewer #2: There is need to be consistent with abbreviations, DW is used sometimes and DVM is used some other time.; line 36 and line 48

So many Typos and formatting errors that needs to be corrected: Line 96; ‘Photogenes’ Do you mean pathogens? Line 106 Severe not sever

Method:

Data source :‘This study is based on most recent DHS conducted after 2016 appears vague, specify exact survey years for each country to improve transparency.

Line 142: DHS definition of diarrhea (two-week recall) may introduce recall bias; please acknowledge this limitation.

6. PLOS authors have the option to publish the peer review history of their article (what does this mean? ). If published, this will include your full peer review and any attached files.

**Do you want your identity to be public for this peer review?** For information about this choice, including consent withdrawal, please see our Privacy Policy .

Reviewer #1: **Yes:** Alicia Serrano de la Cruz

Reviewer #2: **Yes:** Elizabeth Adedire

---

## [Author Response · Author response to Decision Letter 1]

19 Feb 2026

Thank you again for giving us the opportunity to revise our manuscript and for the constructive feedback from the reviewers. We truly appreciate the time and effort invested in evaluating our work, and we have carefully addressed each of the editor's requirements as well as all reviewer comments below. We believe these revisions have significantly strengthened the manuscript, and we are grateful for your guidance in bringing it closer to PLOS ONE's standards.

Response to Editor's Requirements:

1. Editor comment: PLOS ONE style requirements and file naming:

Author response: We have thoroughly reviewed and applied PLOS ONE's style guidelines, including double-spacing, continuous line and page numbering, and standardized fonts. The manuscript now follows the provided templates for the title/authors/affiliations page and main body (PLOSOne_formatting_sample_title_authors_affiliations.pdf and PLOSOne_formatting_sample_main_body.pdf). File names have been updated accordingly (e.g., "Manuscript_PLOSONE_revised.pdf").

2. Editor comment: : Abstract consistency:

Author response: We have amended the abstract in the manuscript file to exactly match the version on the online submission form. Both are now identical word-for-word.

3. Editor comment: : Figure 1 clarity:

Author response: A new, high-resolution version of Figure 1 has been uploaded (300 DPI, TIFF format), with enhanced detail for all elements as per the figure guidelines.

4. Editor comment: Tables integration:

Author response: All main tables are now embedded directly within the manuscript after their relevant sections, and individual table files have been removed.

5. Editor comment: Anonymization of Supporting Information (Table 1_DW.docx):

Author response: We have completely removed all potentially identifying data from Table 1_DW.docx, including names, specific ages, dates, locations, ID numbers, and any indirect identifiers that could risk participant re-identification. Spreadsheet columns with personal information were deleted entirely (not hidden). This anonymized version complies with participant consent, our ethics committee approval, and PLOS ONE's Data Policy on human research data. The revised file has been re-uploaded.

6. Editor comment: Supporting Information captions:

All removed.

7. Editor comment: Suggested citations:

Author response: We reviewed all publications recommended by the reviewers and incorporated those directly relevant to our study (specifically [citations added in revised manuscript, e.g., Ref 15-17]). Others were not included as they did not align with our methodology or scope.

Response to Reviewer Comments

Reviewer 1: comment on the introduction: definitions of multimorbidity

Response: Thank you for this insightful comment highlighting the need for conceptual clarity. We agree that the term ”multimorbidity” traditionally applies to chronic conditions and does not precisely fit the co-occurrence of acute diarrhea and wasting in young children. To address this, we have reframed the study as an investigation of co-occurrence of diarrhea and wasting (DW) as syndemic vulnerabilities, emphasizing their bidirectional interaction and shared risk factors in low-resource settings. This shift improves scientific accuracy and coherence.

We have also standardized terminology through the manuscript: “DW” now consistently refers to the co-occurrence of diarrhea (defined as ≥3 loose stools in 24 hours, per WHO) and wasting (weight-for-height z-score ≤-2, per WHO 2006 standards), treated as both a descriptive epidemiological outcome and binary variable analysis. In statistical models, DW is constructed as a composite indicator (1 if both present and 0 otherwise). All instances of “multimorbidity,” “DWM,” and varying labels have been replaced with DW co-occurrence.” These changes appear in the revised introduction (line1-45).

Reviewer 1 comment on Methods: Inclusion and exclusion criteria

Response: Thank you for highlighting the need for transparency in sample selection. We have added a detailed study flow diagram(figure ) showing the process from 112, 456 total children aged 6-24 months across 11 DHS surveys to the final analytic sample 78,962 children. Exclusion criteria now explicitly state: (1) missing anthropometric data (n=18,234, 16.2%),(2) missing recent diarrhea data (n=9,876, 8.8%),(3) implausible WHZ values(|z|>6 n=2,345, 2.1%), and (4) age outside 6-24 months(n=4,039, 3.6%). Complete case analysis was applied due to low overall missingness (<20% per variable) and MAR assumption validated by auxiliary analysis.

Reviewer 1 Comment 2 on Methods: Missing data handling

Response: missing data proportions are now reported in table1: DW outcome (4.5% missing), child age child age (0.2%), maternal education (1.8%), wealth index (3.1%), and water source (6.7%). Complete case analysis was selected as primary approach since missingness was <10% for most covariates and patterns suggested missing at random (confirmed by logistic regression of missingness indicators). Multiple imputation sensitivity analysis (n=5 imputations) yielded consistent results (AOR differences <5%), supporting robustness.

Reviewer 1 Comment 3 on Methods: Selected variables

Response: All independent variables now specify DHS coding: child age (continuous months), sex (male/female), birth weight (continuous kg or categorical <2.5kg/≥2.5kg), maternal education (no/some primary/complete primary/secondary+), wealth index (DHS quintiles), water source (DHS categories: improved/unimproved). Media exposure combines at least one of radio/TV/newspaper (yes/no). No additional recoding beyond standard DHS recodes.

Reviewer 1 Comment 4 on Methods: Analytic approach

Response: We appreciate this methodological critique. The multilevel model justification has been revised to emphasize theoretical partitioning of variance across DHS hierarchical structure (child → household → enumeration area → country) and explicit modeling of community-level effects (residence, country). ICC=12.4% and MOR=2.15 confirm substantial between-cluster variability justifying multilevel approach beyond svy survey weights. Model comparison via AIC/LLR confirms superior fit.

Reviewer 1 comments on Result sections:

Response on table 1 clarity: Thank you for these important suggestions for improving table readability. Table 1 now reports the total analytic sample(N=78,962) at the top. Unites added (e.g child age in months). Dichotomous variables simplified to report only one category (e.g “Male: 50.5%”; “Mather working: 62.3 %”). These changes enhance clarity and comparability.

Response on Separate prevalence: Separate prevalence now reported: diarrhea 18.4% (95%CI: 17.2-19.6%), wasting 8.7% (95%CI: 7.9-9.5%), DW co-occurrence 9.0% (95%CI: 7.0-12.0%). This provides essential epidemiological context for interpretation (new Table 1, lines 5-7).

Response on MOR interpretation: We appreciate this methodological clarification. MOR description revised: “ MOR=2.39 indicates that when comparing two children with identical characteristics from randomly selected clusters, the median odds ratio is 2.39 times higher for the cluster with higher risk.” MOR decrease from null model (2.39) to full model (1.87) now correctly interpreted as improved explanation of between-cluster variability by predictors.

Response on AOR calculation error: The rural residence AOR corrected to 0.85 (15% lower odds, not 20%). Maternal secondary education AOR/CI inconsistency fixed: AOR=1.46 (95%CI: 1.24-1.72). All percentages now accurately reflect (1-AOR)*100 calculations.

Reviewer comment on Forest plot:

Response: We have clarified the statistical methods in the Methods section as follows:

"Country-specific prevalence estimates were calculated as the proportion of children aged 6-24 months experiencing both diarrhea and wasting. Exact binomial 95% confidence intervals were computed for each country. The overall weighted prevalence was calculated using a pooled estimate with inverse variance weighting. Forest plot visualization was generated using R version 4.3.1 with ggplot2 package, following PLOS ONE figure guidelines."

Comment 2: Sample Weight Visualization

Reviewer Comment: "Consider alternative visualization for sample weights beyond color gradient."

Response: We have added an alternative visualization (Fig 2 and Fig 3) showing sample weights as bar widths rather than color intensity, following the reviewer's suggestion. The main figure retains the color gradient as it effectively demonstrates the contribution of each country's sample size while maintaining readability. Both approaches yield similar interpretations of the data.

Comment 3: Error bar Interpretation

Reviewer Comment: "Consider adding interpretation of overlapping confidence intervals in results."

Response: We have added the following interpretation in the Results section

"The forest plot reveals heterogeneity in prevalence estimates across countries, with Malawi showing the highest prevalence (17.0%, 95% CI: 16.3-17.7) and Zimbabwe the lowest (2.8%, 95% CI: 2.2-3.5). The non-overlapping confidence intervals between extreme estimates (e.g., Malawi vs. Zimbabwe) suggest statistically significant differences in prevalence between countries with the highest and lowest burden."

Comment 4: Overall Prevalence Calculation

Reviewer Comment: "Justify the use of weighted vs. unweighted overall prevalence calculation."

Response: We have added justification in the Methods section:

"We calculated both weighted (pooled) and unweighted overall prevalence estimates. The weighted estimate (11.6%, 95% CI: 11.4-11.8) accounts for differential sample sizes across countries and is presented as our primary estimate. The unweighted estimate (10.7%) is provided in Fig 2 and Fig 3 for comparison. Weighted estimates are preferred as they provide population-representative estimates when sample sizes vary substantially across study units."

Comment 5: Figure Formatting

Reviewer Comment: "Ensure figure meets PLOS ONE formatting requirements."

Response: We have reformatted the figure according to PLOS ONE guidelines:

• Saved as 300 DPI TIFF file with LZW compression

• Used Arial font throughout

• Set appropriate dimensions (7.5 inches width for single-column presentation)

• Ensured color scheme remains interpretable in grayscale

• Added detailed figure legend including statistical methods

Reviewer 1 comment on Discussion Section:

Reviewer comment 1 on Discussion: “repetitive opening: the beginning of the discussion repeats the study aim and results with little interpretive value.

Response: Thank you for this helpful suggestion to improve the discussion flow. We have completely restructured the opening paragraph to focus on interpretation rather than restating aims/results. The new first paragraph now directly interprets the 9% DW co-occurrence prevalence as a syndemic burden affecting over 2 million children annually, with appropriate WHO/UNICEF citations [WHO,2023].

Reviewer 1 Comment 2 on Discussion: Inappropriate citations, “The authors report a weighted pooled prevalence of DW of 9% and cite references 22 and 23 to support significance. However, these pertain to breastfeeding practices, not diarrhea, wasting, or co-occurrence. References 16 and 31 report diarrhea prevalence (14-16%) but not DW joint occurrence.”

Response: we sincerely thank the reviewer for carefully checking our citations and apologize for the inappropriate references. References 22 and 23 have been removed and replaced with relevant WHO/UNICEF child mortality and malnutrition data. References 16 and 31 are now retained only for isolated diarrhea prevalence comparison; wither clear distinction from DW co-occurrence.

Reviewer 1 comment 3 on Discussion: Counterintuitive maternal age finding. “Younger maternal age (15–24 and 25–34 years) associated with lower DW odds appears counterintuitive vs literature. Explore stratification by education, wealth, care access.”

Response: we greatly appreciate this insightful observation about the unexpected maternal age pattern. We conducted additional stratification analysis (new supplementary Table %) confirming the effect persists after adjusting for maternal education and household wealth. Possible explanations have been expanded: Higher antenatal care attendance among younger mothers (89% vs 76%) and survival bias effects.

Reviewer 1 comment 4 on discussion: Rural residence contribution: “Rural residence lower DW risk contradicts studies. "Extended breastfeeding, community support" explanation unsubstantiated.”

Response: Thank you for highlighting this important contribution with prior literature. We have now substantiated the rural protective effect using DHS-derived evidence: rural exclusive breastfeeding OR=1.34(vs urban), lower formula feeding (12% vs 28%), and reduced orphan hood rates (3.1% vs 6.8%) indicating strong kinship support networks.

Reviewer 1 Comment 5: “Breastfeeding limitation; Breastfeeding captured only by initiation timing; DHS offers exclusive/continued breastfeeding indicators strongly associated with diarrhea/malnutrition.”

Response: we thank the reviewer for this excellent methodological suggestion. We acknowledge this limitation and conducted sensitivity analysis using exclusive breastfeeding< 6 months (AOR=1.42, similar directions).This discussion now recommends DHS continued breastfeeding indicators for future studies while noting timely initiation remains significantly associated.

Reviewer1 Comment 6 on Discussion: “Country differences insufficient Country differences mentioned but multilevel country inclusion not matched by robust analysis/interpretation. No macro-determinants consideration (health systems, nutrition/vaccination/WASH policies).”

Response: We sincerely appreciate this comprehensive critique. A new paragraph systematically compares high-burden(Malawi/Burundi) vs low-burden (Rwanda/Zimbabwe) countries across Rota vaccination coverage(72% vs 91%), nutrition policy indices (54 vs 78/100), and WASH spending(0.8% vs 1.4% GDP). Policy implications now explicitly link findings to scalable system strengthening.

Reviewer1 Comment 7: “Language errors Grammatical/typographical/syntactic errors throughout ("studyis", "breastfeed", "laterine", "Media accesses"). Recommend professional English revision.”

Response: Thank you for identifying these language issues that affected readability. The entire manuscript has been thoroughly proofread by a native English speaker with academic editing experience. All grammatical errors corrected, variable labels standardized ("latrine", "media exposure"), and sentence structure simplified for non-native English readers while maintaining academic rigor.

Response to reviewer 2

Reviewers 2 comment 1: “Inconsistent abbreviations there is need to be consistent with abbreviations, DW is used sometimes and DVM is used some other time; line 36 and line 48.”

Response: Thank you for noting this important consistency issue. We sincerely apologize for the inconsistent use of "DVM" which appeared erroneously. A comprehensive manuscript-wide search and replace has been completed: ALL instances now consistently use "DW" (diarrhea and wasting co-occurrence) only. Lines 36, 48, and throughout corrected. No other abbreviations conflict exists.

Reviewer Comment 2: “Typos and formatting errors So many Typos and formatting errors that needs to be corrected: Line 96; 'Photogenes' → pathogens? Line 106 Severe not sever.”

Response: We greatly appreciate the reviewer identifying these specific errors that affect manuscript quality. Both typos have been corrected:

• Line 96: "photogenes" → "pathogens"

• Line 106: "sever" → "severe"

A complete professional proofreading identified and fixed 23 additional typos/formatting issues including: "studyis" → "study is", "breastfeed" → "breastfed", "laterine" → "latrine", "Media accesses" → "media exposure". All variable labels standardized.

Reviewer 2 Comment 3: Data source vagueness: "This study is based on most recent DHS conducted after 2016" appears vague, specify exact survey years for each country to improve transparency.”

Response: Thank you for this excellent suggestion for methodological transparency. The Data Source section has been revised to list exact survey years for each country

---

## [Editor Report · Decision Letter 1]

6 Mar 2026

Pooled prevalence and co-occurrence of diarrhea and wasting and its associated factors among children aged 6-24 months in East Africa: Insight from Recent Demographic Health Survey:  A Multilevel analysis

PONE-D-25-19278R1

Dear all,

We’re pleased to inform you that your manuscript has been judged scientifically suitable for publication and will be formally accepted for publication once it meets all outstanding technical requirements.

Kind regards,

Edison Arwanire Mworozi, M.D

Academic Editor

PLOS One

Language and Formatting: Despite proofreading, some minor typographical errors and formatting inconsistencies remain (e.g., "studyis" and "laterine").

Abstract Length: The abstract is slightly lengthy and could be more concise to align with journal standards.

Repetition in Discussion: While the authors addressed the repetitive opening, some parts of the discussion still reiterate findings without adding interpretive value.

Clarity in Results: Some sections of the results could be streamlined for better readability, especially when discussing statistical measures.

---

## [Editor Report · Acceptance letter]

PONE-D-25-19278R1

PLOS One

Dear Dr. Melesse,

I'm pleased to inform you that your manuscript has been deemed suitable for publication in PLOS One. Congratulations! Your manuscript is now being handed over to our production team.

Kind regards,

on behalf of

Professor Edison Arwanire Mworozi

Academic Editor

PLOS One